# Mechanical and Ballistic Properties of Epoxy Composites Reinforced with Babassu Fibers (*Attalea speciosa*)

**DOI:** 10.3390/polym16070913

**Published:** 2024-03-26

**Authors:** Yago Soares Chaves, Sergio Neves Monteiro, Lucio Fabio Cassiano Nascimento, Teresa Gómez-del Rio

**Affiliations:** 1Department of Materials Science, Military Institute of Engineering-IME, Praça General Tíburcio, 80, Urca, Rio de Janeiro 222290-270, Brazil; snevesmonteiro@gmail.com (S.N.M.); lucio@ime.eb.br (L.F.C.N.); 2Durability and Mechanical Integrity of Structural Materials Group (DIMME), School of Experimental Science and Technology, Rey Juan Carlos University, Mostoles, 28933 Madrid, Spain; mariateresa.gomez@urjc.es

**Keywords:** babassu fiber, composite, tensile properties, Izod, ballistic properties, failure mechanisms

## Abstract

The mechanical and ballistic performance of epoxy matrix composites reinforced with 10, 20, and 30 vol.% of babassu fibers was investigated for the first time. The tests included tension, impact, and ballistic testing with 0.22 caliber ammunition. The results showed an improvement in tensile strength, elastic modulus, and elongation with the addition of babassu fiber, and the 30 vol.% composite stood out. Scanning electron microscopy analysis revealed the fracture modes of the composites, highlighting brittle fractures in the epoxy matrix, as well as other mechanisms such as fiber breakage and delamination in the fiber composites. Izod impact tests also showed improvement with increasing babassu fiber content. In ballistic tests, there was an increase in absorbed energy. All composites surpassed plain epoxy by over 3.5 times in ballistic energy absorption, underscoring the potential of babassu fiber in engineering and defense applications.

## 1. Introduction

The ballistic protection materials that are currently in use are an evolution of the old forms of combatant protection used to keep up with the advance of the weapons used in armed conflicts [1]. Current ballistic materials have been developed for the purpose of providing individual protection. These materials have properties suitable for the wearer. Steel or aluminum alloys provide effective protection against high-caliber ammunition. However, a significant disadvantage of these materials is their relatively high density, which can compromise mobility. For this reason, they are more suitable for use in vehicles than for individual protection [2,3,4].

Composite materials are widely used owing to their superior mechanical properties compared to conventional materials. The matrix phase, in turn, is responsible for involving and protecting the reinforcement phase, besides providing strength and stiffness to the composite [5,6]. 

Synthetic fibers, such as fiberglass, carbon, and aramid, are commonly employed as reinforcement phases due to their high strength and stiffness. These fibers confer composite properties such as high tensile strength, but these materials also present relatively high density compared to other materials [7,8]. On the other hand, natural fibers, such as hemp [9], jute [10], and bamboo [11], have gained prominence due to their low density as well as their renewable and sustainable properties. These fibers are widely used in automotive applications [12], construction [13], and ballistic armor [5], providing environmental benefits by replacing conventional petroleum-based materials and being found in abundance in nature [14,15].

In an event involving the impact of a high-velocity projectile against a fiber-containing polymeric plate, the material develops a distinct fracture mechanism, such as delamination, matrix crack pattern, fiber breakage, and pull-out [16]. Certain mechanisms play an extremely important role in impact energy absorption in composite materials. However, there are other mechanisms, such as delamination, that can compromise the integrity of the target after the first ballistic shot. This phenomenon results in the loss of its ability to protect against subsequent shots, as required by regulatory standards for ballistic testing [3]. The research into the ballistic properties of various natural fibers is driven by the recognition that each of them possesses unique characteristics and properties. Although they share some similar components, such as lignin and cellulose, the distinct composition and characteristics, such as crystallinity index, fiber diameter, and microfibril angle, exert a direct influence on the mechanical properties and ballistic resistance of composites reinforced with this type of material [17,18].

When considering the use of natural fibers for ballistic applications, it is evident that numerous studies explore the potential of fibers such as jute, ramie, sisal, and curaua, among others [19,20,21]. However, the use of Amazonian fibers, not only for this specific purpose but also for engineering in general, remains largely unknown. The Amazon region stands out for its extraordinary diversity, with numerous native plants providing natural fibers of excellent mechanical quality [22]. A notable example is the babassu fiber. Extracted from the babassu palm tree (*Attalea speciosa*), this fiber belongs to the Attalea genus of the Arecaceae family and is obtained from the tree’s nuts. Babassu palms are widely distributed in a vast region of Brazil, spanning from the Amazon to the Cerrado, encompassing a rich diversity of vegetation [23]. Babassu fiber stands out for its low cost, as it is manually collected in regions close to the vegetation. Additionally, it exhibits excellent mechanical properties, as evidenced in Table 1. Although babassu fiber is widely used in various applications, from animal feed to use in soils and thermoplastic polymer compounds [24,25], it is important to note that composite materials reinforced with babassu fiber have not yet been extensively studied regarding their mechanical and ballistic properties.

Given the relevance of studying natural fibers for ballistic applications, the aim of this article is to investigate the mechanical and ballistic properties of babassu fiber-reinforced epoxy composites. Understanding the mechanical properties of these composites will not only enable their use in ballistic applications but also in engineering composites in general. The composites analyzed in this study underwent tensile, Izod impact, scanning electron microscopy, and ballistic testing with a 0.22 caliber projectile. Thus, we present an innovative work in the literature, introducing a previously unexplored composite.

## 2. Materials and Methods

### 2.1. Materials

In the manufacture of the fiber–polymer matrix plates, the epoxy resin DGEBA was used together with a triethylenetetramine (TETA) based hardener, maintaining a stoichiometric ratio of 100 parts of epoxy to 13 parts of TETA. The materials used were supplied by Dow Chemical, located in São Paulo, and distributed by Epoxy fiber, a company based in Rio de Janeiro, Brazil.

The babassu fibers were extracted from the nuts coming from babassu palm trees in the city of Santana do Maranhão (MA, Brazil). The process of extracting the babassu nut fibers involved several careful steps. First, the nuts were broken after being exposed to the sun for a certain period of time. Next, the nuts were submerged in water for 7 days in order to improve pliability for subsequent defibrillation. After being removed from the water, the nuts were dried in an oven at a fixed temperature of 80 °C for 24 h. After the drying period, the manual defibrillation process of the babassu nut fibers was performed, resulting in babassu fibers [26].

### 2.2. Sample Preparation

The babassu fiber–matrix plates were produced by randomly arranging the fiber in a metal mold with the dimensions 150 × 120 × 12 mm, as shown in Figure 1.

After the drying process in an oven at 80 °C for 24 h, the babassu fiber is transferred to a metal mold that was previously lubricated all over its surface with silicone grease. The fiber is carefully placed in the mold and the resin is spread so as to cover its entire length, ensuring adequate wettability and adhesion between the fiber and the matrix. The plate was obtained after hydraulic thinking under a load of 5 tons for a period of 24 h, using a hydraulic press (Skay Industry, São Paulo, Brazil). The average diameter found for babassu fibers was 0.25 µm [26]. 

To determine the mixture proportion between the epoxy resin and hardener, the rule of mixture method was used, and the density of babassu fiber was found to be 0.79 ± 0.03 g/cm^3^ [26]. Regarding the epoxy resin, the density of 1.11 g/cm^3^ is in accordance with literature data [9]. Epoxy composites with 10, 20, and 30 vol.% of babassu fibers were manufactured for mechanical and ballistic tests. In Table 2, the nomenclature of each sample group is presented along with the content of epoxy resin and babassu fiber used.

### 2.3. Ballistics Test

The ballistics tests were conducted using an SSS and Gunpower model compressed air rifle (Figure 2), equipped with two communicating cylinders (0.5 L and 6 L) and an estimated pressure of 28 MPa. The projectile used was a 0.22 caliber lead (Figure 2b) measuring 1 cm in diameter and weighing 14.4 g. The ballistic test schematic is shown in Figure 3.

To determine the absorption energy, two ProChrono Pal model ballistic chronographs with an accuracy of 0.31 m/s were employed to measure both impact velocity and residual velocity. The distance between the air rifle and the composite plate was 5 m. The samples were placed on a metal stand, as illustrated in Figure 3. The wooden plate was used to retain the velocity of the projectile in order to analyze the impact resistance. The energy absorbed by the composite plate is calculated using Equation (1):(1)$$E_{abs}=\;\frac{m_{p}({v^{2}}_{i}-\;{v^{2}}_{r})}{2}$$
where E_abs_ is the energy absorbed by the compound (J), m_p_ is the mass of the projectile (g), v_i_ is the impact velocity, and v_r_ is the residual velocity (m/s).

Another crucial parameter is the limit velocity (V_L_), which can be determined by manipulating Equation (1) when all the kinetic energy of the projectile is absorbed and the residual velocity is reduced to zero. This relationship can be formalized by means of Equation (2), where a mathematical model is established to express this condition:(2)$$V_{L}=\sqrt{\frac{2E_{abs}}{m}}$$

### 2.4. Mechanical Tests

#### 2.4.1. Tensile Tests

Tensile testing was conducted using an INSTRON 3365 universal machine, with load cell parameters of 10 KN and a tensile rate of 2 mm/min. Samples were obtained from epoxy/babassu composite plates (120 × 15 × 12 mm) and were cut into seven specimens following a procedure adapted from ASTM D3039 [28]. From the tensile test, the values of tensile strength (σ), elastic modulus (E), and total strain (ε) were obtained.

#### 2.4.2. Izod Impact Test

The Izod impact test was conducted according to ASTM D 256-10 [29] using Panantec ATMI equipment, São Paulo, Brazil, with a 22 J hammer. For the statistical feasibility of the tests, seven specimens with dimensions of 63.5 × 12.7 × 10 mm were produced. Notching was performed manually with a notcher, with a depth of 2.54 mm and an angle of 45° [30].

### 2.5. Scanning Electron Microscopy (SEM)

The SEM analysis associated with the main failure mechanisms on the fracture surfaces that occur in the composites was performed after the tensile and ballistic tests. A Quanta FEG 250 Fei microscope, Hillsboro, OR, USA, equipped with an Everhart–Thornley secondary electron detector was used, operating at an acclimation voltage of 10 kV. In order to visualize the flaws in the composite, the samples were coated with gold using a Leica ACE600 sputtering machine, São Paulo, Brazil.

### 2.6. Statistical Analysis

The results of the ballistic tests were subjected to statistical analysis using variance analysis (ANOVA), with a confidence level of 95% for all tests. It also provided an assessment of whether the number of fibers used as reinforcement of the composites influences the results obtained. In order to verify the presence of significant differences between the mean values of the results, Tukey’s test, also known as the honest significant difference (HSD) test, was used, as calculated by Equation (3).
(3)$$HSD=q\sqrt{\frac{EMS}{r}}$$
where q = HSD constant tabulated for 5% significance; MSE = root mean square error of ANOVA; r = number of replicates for each treatment [31].

## 3. Results and Discussion

### 3.1. Ballistic Tests

The ballistic test results were obtained using Equations (1) and (2), respectively. The energy absorption (E_abs_), residual velocity (V_r_), and limit velocity (V_L_) values for each sample are presented in Table 3. It can be seen from the results that several factors influence the performance of each sample, such as the percentage of fibers used as reinforcement. However, the reliability of the results depends on statistical analysis, in particular the Weibull Analysis.

Before proceeding to the analysis of the values obtained for the energy absorbed in the ballistic test, the data were subjected to the ANOVA test. This analysis allowed determining the reliability of the samples, which vary from 0 to 30 vol.% in the fiber content. The results of this analysis are presented in Table 4.

At each level of fiber concentration present in the composite, the variance test (Table 5) evaluated the dispersion of the data in relation to the mean obtained. This indicator is fundamental to evaluating the consistency of the results achieved [32].

With the variance result obtained (Table 5), it is necessary to calculate the F value, which is defined by comparing the variance in the calculations. This parameter is essential to assess the significance of differences between group means in the ANOVA [33]. The values of calculated F, *p*-value, and critical F are presented in Table 6.

The results of the statistical analysis performed shown in Table 6 reveal that the calculated F value is significantly higher, and the *p*-value is significantly lower, which suggests greater relevance of the parameters within the 95% confidence range. The low *p*-value obtained (0.21) indicates statistical significance [34]. The presence of different amounts of fibers in the composites directly impacted the E_abs_, and this effect was confirmed by the ANOVA test.

Tukey’s test was used to establish associations between the means of the different samples. The results revealed a value of 7.67 for the three treatments evaluated, with a degree of freedom of 12°. By applying Equation (3), it the HSD value was determined to be 69.92 J [31,35,36]. Table 7 illustrates the relationship between the mean values of each sample, considering the incorporation of babassu fibers.

In the Tukey analysis, Table 6 shows that the means of the 10, 20, and 30 vol.% groups are statistically equal. This means that, within the context of the study, these means do not differ in a statistically significant way. This identification of equal means reinforces the understanding that certain conditions or treatments have similar impacts, as evidenced by the statistical analysis [31,37,38,39].

Figure 4 shows the fractures of the epoxy/babassu samples before and after the ballistic test.

Analyzing Figure 4d,e, the samples EB10 and EB20 did not exhibit a preferentially oriented fracture pattern, which resulted in the total fragmentation of the samples. This behavior is not desirable in ballistic armor contexts, where it is crucial that the material can withstand multiple gun shots. However, the sample EB30 (Figure 4f) showed more robust behavior, maintaining its integrity even after being subjected to five rounds of ammunition from the 0.22 caliber weapon, without showing visible signs of rupture. This points to a more promising condition in terms of applicability in ballistic protection scenarios.

Considering the use of composite plates as ballistic armor, subject to multiple gun shots, preserving the physical integrity of the material is of great importance in the process of selecting the right material. In this context, the EB30 composite is the most suitable choice. A determining factor lies in the fact that increasing the proportion of fibers leads to a corresponding reduction in the amount of resin used, which has a positive impact on reducing the final cost of the composite. In addition, analysis of the mean values using Tukey’s statistical test shows a remarkable similarity between the means, reinforcing the viability and consistency of choosing the sample with 30 vol.% babassu content.

The complete fragmentation of the EB10 and EB20 composites can be attributed to the brittle characteristics of their epoxy matrix. This is because the epoxy matrix plays a crucial role in dissipating the energy resulting from the impact of the ammunition on the composite. In addition, the amount of reinforcement may also be one of the underlying reasons for fragmentation, since the matrix will have difficulty transferring loads to the reinforcement containing the fibers. This is evidenced by the analysis of the EB30 composite plate, where this load transfer is more effective, resulting in a greater capacity to absorb the energy generated when the ammunition is fired [40,41,42,43]. The failure mechanisms, as well as the inefficiency in energy dissipation due to the absence of fibers in the reinforcement, are clearly evident in the SEM analysis.

Several factors influence the failure mechanism, the epoxy matrix being one of the most important. Its brittle nature contributes to the occurrence of various types of fractures after ballistic tests. The SEM analysis in Figure 5 illustrates these mechanisms, showing river marks that denote the fragility of the epoxy matrix, as well as fiber pull-out resulting from the high-intensity impact. Both under static loads and after dynamic events, composites are subject to a type of failure that is associated with low fiber/matrix interfacial adhesion [44].

### 3.2. Mechanical Properties

#### 3.2.1. Tensile Tests

Table 8 and Figure 6 show the results of the tensile tests for the different contents of babassu fibers present in the composite samples.

The values obtained in Table 8 and Figure 6 are consistent with the literature compared to the results of Sature and Maché [45], who carried out mechanical characterization and water absorption studies on hybrid composites reinforced with jute and hemp, with a volume fraction of 40% of reinforcing fibers. The authors obtained 58.03 MPa for the epoxy composite with jute fiber and 75.14 MPa for the epoxy composite with hemp fiber. 

In the study conducted by Shah et al. [46], the results of a composite using an epoxy matrix and flax and sisal fibers as reinforcing materials were documented. This composite had a maximum tensile strength of 39.22 MPa and Young’s modulus of 2.37 GPa. On the other hand, Kalagi et al. [47] reported a composite using an epoxy matrix and jute and coconut fibers as reinforcement, achieving tensile strength values of 37.5 MPa and Young’s modulus of 1.5 GPa.

In addition, the study conducted by Lau et al. [48] provided a comparative analysis of composites using an epoxy matrix with hemp fibers and a similar matrix with jute fibers. The results showed that the hemp composite had a maximum tensile strength of 75 MPa and Young’s modulus of 3.33 GPa, while the jute composite had values of 58 MPa for tensile strength and 4 GPa for Young’s modulus. In the case of the composite containing jute, the results are comparable to those obtained with 30% babassu fibers, with a value of 46.13 ± 6.87 MPa. 

An important factor in the processing of fiber-reinforced composites is alignment. The orientation of fibers plays a significant role in mechanical strength and energy absorption. An interesting example is the paper by Rong et al. [49], where the authors investigated two crucial factors in the use of natural fibers as reinforcement in epoxy matrix composites: the surface treatment of fibers and a high concentration of fibers as reinforcement, along with the use of alkaline treatment on the fibers. Through tensile test results, the authors, using composites reinforced with 77% aligned unidirectional sisal fibers, achieved a tensile strength of 330 MPa and Young’s modulus of 6.00 GPa. This result, which is considerably higher than the results obtained for epoxy/babassu composite samples, demonstrates the importance of fiber alignment.

In the case of studies exploring the use of babassu fibers as reinforcement in composites, Moura et al. [50] investigated babassu composites with the addition of the plasticizer glycerol. They obtained a tensile strength of 31.67 ± 1.07 MPa, Young’s modulus of approximately 0.12 GPa, and an elongation of 10.63 ± 1.6%. It is worth noting that the results obtained for the babassu reinforcement are lower than those of the present study. This difference can be attributed to the type of application of the fiber since glycerol and epoxy resin have substantially different properties.

As might be expected, since fibers generally have greater strength and stiffness compared to the matrix, it is common for the strength and stiffness of the composite to increase as the proportion of fibers is increased. However, this is conditional based on adequate interfacial resistance between the fiber and the matrix, as this resistance can be reduced in cases involving strongly hydrophobic matrices [51].

As was performed in the analysis of the ballistic results, it is essential to conduct a statistical analysis to determine the F value. By applying ANOVA, the results of the tensile strength tests of the babassu fibers were statistically analyzed in order to achieve a confidence level of over 95% [52,53]. In addition, this analysis made it possible to assess whether the number of fibers used as reinforcement in the composites had an influence on the results obtained [54,55]. Table 9 shows the results for tensile strength, modulus of elasticity, and deformation subjected to statistical analysis using the ANOVA test.

As observed in the properties of tensile strength, Young’s modulus, and total deformation, the F values, calculated as (8.56), (27.95), and (10.60), respectively, are higher than the critical F value found (2.95). Therefore, the null hypothesis of statistically significant differences between the properties of the different types at a 95% confidence level is rejected. To confirm the ANOVA results, Tukey’s test (Table 10) was carried out for tensile mechanical properties.

The HSD obtained for the properties of tensile strength, Young’s modulus, and deformation were (9.05), (32.63), and (28.19), respectively. These results indicate which treatment shows a difference in their average values. When the mean values of the groups compared two by two are greater than the HSD, it is considered that there is a difference between the mean values evaluated. For the tensile strength values, Table 9 shows that the values differ with the increase in fiber percentage, whereas for Young’s modulus and deformation, there is no difference with the increase in fiber percentage, and the value does not differ at a 95% confidence level.

Figure 7 shows the images obtained by SEM that reveal the fracture surface of the composites, as well as the different rupture modes that occurred during the tensile test.

The analysis of the fracture surface, shown in Figure 7, of the tensile-tested composites shows the predominant failure mode. In addition, it is possible to observe the occurrence of rupture in interface regions between the fiber and the matrix, as illustrated in Figure 5a. When composites are subjected to static loads or after dynamic events, a type of failure occurs which is characterized as a mechanism of low interfacial adhesion between the fiber and the matrix [44]. Additionally, Figure 7b reveals river marks, which are common failure mechanisms in the matrix, as well as an imprint of the fiber on the matrix surface. This feature indicates a significant level of adhesion between the fiber and the matrix [54].

#### 3.2.2. Izod Impact Test

Table 11 and Figure 8 show the results obtained for Izod impact tests of epoxy matrix composites reinforced with babassu fiber as well as pure epoxy resin.

The main objective of the impact test is to determine the toughness of the material and its dynamic ability to absorb energy. This ability to withstand dynamic loads is known as impact resistance [56]. Table 10 compares the notched Izod impact resistance between the pure epoxy resin sample and the epoxy/babassu composites. In comparison to EB10 and EB20 composites, the sample EB30 showed a higher energy absorption capacity, reaching 62.53 ± 7.43 J/m. This value represents an increase of 17% compared to the EB20 composite. These findings indicate that an increase in the percentage of fibers in the composites results in a greater capacity to absorb impact energy.

The energy absorption results obtained by the epoxy matrix composite containing babassu fiber revealed a higher value than is normally found in the literature [57,58,59,60]. However, this difference is subject to variations due to the specific characteristics of the fibers, which vary from one natural fiber to another. This variation emphasizes the notable discrepancies between the different types of natural fibers documented in the literature [30,61,62,63,64,65,66,67]. ANOVA was also used to examine the variations in the resistance to energy absorption in the Izod impact test, as shown in Table 12.

Based on the results of the ANOVA test shown in Table 11, the hypothesis that the mean value is equal is rejected at a 5% significance level. In fact, the statistical test indicated that the calculated value of F (47.46) is greater than the critical value of F (tabulated) (2.95). Therefore, one may conclude that the volumetric content of babassu fibers in epoxy matrix composites indeed has a significant impact on the Izod impact energy. In addition, Tukey’s test (Table 13) was conducted to compare the means, with a 95% confidence level, in order to determine which volumetric content of babassu fibers provides the best results in terms of Izod impact energy.

Based on the data presented in Table 13, the calculated HSD was 9.25. Therefore, differences greater than this value were considered statistically significant, indicating that at least 10 vol.% babassu fibers must be incorporated to obtain an effective reinforcement in epoxy/babassu composites. There were no significant differences in the averages between the epoxy and EB10 samples, whereas EB20 and EB30 showed statistically significant differences in relation to the average, as shown in Table 13.

The increase in impact energy as the amount of babassu fibers in the composite increases may be related to the different fracture mechanisms in composites with different percentages of babassu fiber. To confirm and deepen the understanding of the evolution of the fracture mechanisms in the materials tested, the SEM images of the fracture surfaces of the samples are shown in Figure 9.

In Figure 9a, it is possible to identify the fracture mechanisms acting on the composites containing babassu fibers, including the brittle fracture mechanism, which manifests itself in the “river marks” shown in the micrographs obtained from the composites. Another mechanism observed is the rupture of the fibers incorporated into the matrix due to the impact of the Izod equipment’s pendulum, a common feature in the micrographs analyzed. In addition, there is a rupture in the adhesion between the fiber and the matrix due to the high energy generated in the test. In Figure 9b, it is possible to identify the occurrence of “pull-out”, a result of the considerable load exerted by the Izod impact.

### 3.3. Literature Comparison

Given the innovative nature of the processing and mechanical/ballistic characterization of epoxy/babassu fiber composites, it becomes imperative to conduct a comparative analysis to determine their real-world applicability, both in ballistic vests—as detailed in this study—and in engineering applications, where they can significantly enhance the mechanical properties of materials. Table 14 presents the key properties of these composites, along with data from other natural fiber-reinforced composites, enabling a precise comparison with the existing literature. This establishes a solid foundation for future investigations and potential industrial applications.

The analysis of the data in Table 14 reveals that epoxy/babaçu composites exhibit quite favorable properties, surpassing some natural fiber-reinforced composites mentioned in the literature. Although there is a lack of information on tensile strength under similar test conditions for other composites, it is noteworthy to mention the performance of the EB20 and EB30 composites in this study compared to the results of Oliveira et al. [70], who investigated epoxy matrix composites reinforced with 40 vol.% of tucum fibers. The composites in this study demonstrated superior values. However, it is important to highlight that the results obtained fell short of those achieved by Ribeiro et al. [9], who used fabric instead of isolated fibers in the fabrication of epoxy/hemp composites, resulting in higher mechanical strength.

The impact resistance of epoxy/babassu composites was found to be low, even with an increase in the fiber content in the composite. The results were similar to those found in the study by Cunha et al. [68], where the authors used 40 vol.% of titica vine fibers in the epoxy matrix. In comparison, studies with hemp [9], caranan [30], and tucum [70] fibers showed significantly higher impact results. One possible explanation for this could be the arrangement of the fibers within the matrix. In this study, the fibers were randomly arranged, which may have compromised the impact resistance of the composites.

Regarding ballistic resistance against 0.22 mm ammunition, the epoxy/babassu composites showed the highest absorption among the compared studies, as well as the lowest VL values. This indicates that the material has a greater capacity for absorption without rupture, enabling it to withstand a greater number of shots. Additionally, the lower VL value suggests that the bullet had its speed considerably reduced, demonstrating greater ballistic resistance due to the use of babassu fibers. Therefore, for engineering applications, these composites still require further studies, which could improve their properties through chemical treatments. However, for ballistic applications, the combination of babassu fiber with epoxy shows great potential.

## 4. Conclusions

In this study, epoxy matrix composites were processed with the addition of babassu fibers in fractions of 10, 20, and 30 vol.%, and their mechanical and ballistic properties were evaluated through tensile, Izod impact, and ballistic tests using 0.22 mm ammunition. The results indicate that the inclusion of fibers resulted in an increase in tensile strength, with the EB30 composite showing the highest value (46.13 ± 6.87 MPa), although it was still lower than the values found in the literature. Energy absorption, measured by the Izod test, also showed improvements with an increase in the babassu fiber content, reaching 62.53 ± 7.43 J/m. Scanning electron microscopy (SEM) analysis revealed fracture mechanisms in the composites in response to the three types of tests, with the predominance of brittle fracture in the epoxy matrix, exemplified by “river marks”. Other observed failure mechanisms included fiber breakage, rupture at the fiber–matrix interface, as well as pull-out mechanisms.

In ballistic tests, the epoxy matrix composite with 10 vol% of babassu fibers showed the best properties, maintaining plate integrity after residual velocity tests. Other composite samples with a lower addition of babassu fibers showed inferior results due to the brittle behavior of the epoxy and observed failure mechanisms. The epoxy/babassu composites demonstrated superior ballistic resistance performance, emerging as a promising option for this application.

## Figures and Tables

**Figure 1 polymers-16-00913-f001:**
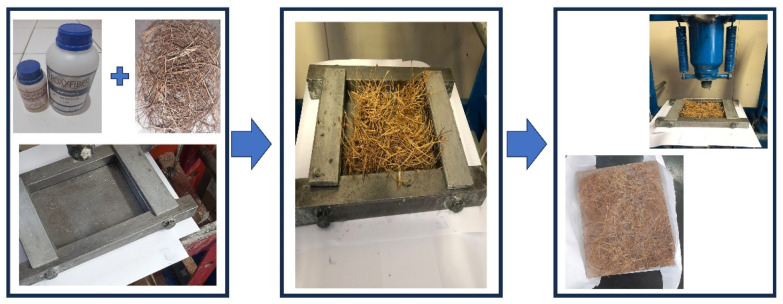
Representation of the fabrication of the babassu fiber–matrix board.

**Figure 2 polymers-16-00913-f002:**
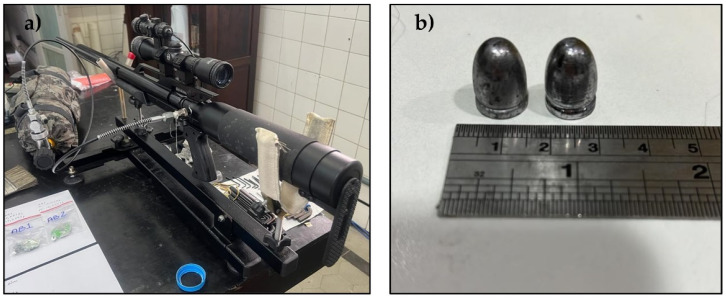
Ballistic test: (**a**) Compressed air rifle model SSS; (**b**) lead projectiles 0.22 caliber.

**Figure 3 polymers-16-00913-f003:**
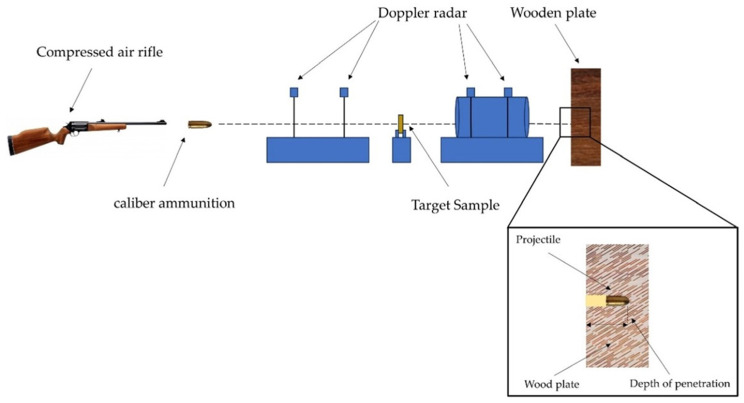
Schematic representation of the ballistic test setup.

**Figure 4 polymers-16-00913-f004:**
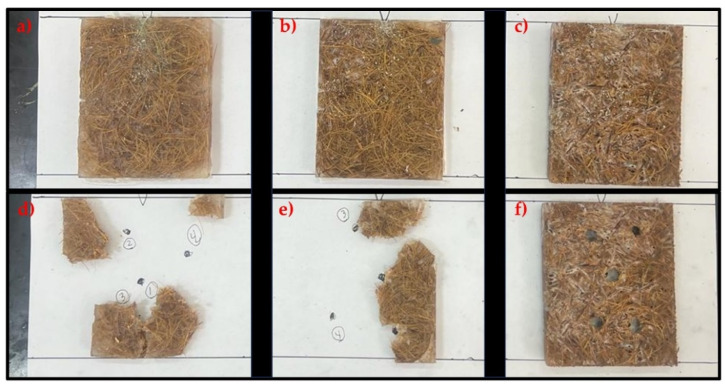
Fractures in composite sample containing 10, 20, and 30 vol.% babassu fiber: (**a**–**c**) before the ballistic test; (**d**–**f**) after the ballistic test.

**Figure 5 polymers-16-00913-f005:**
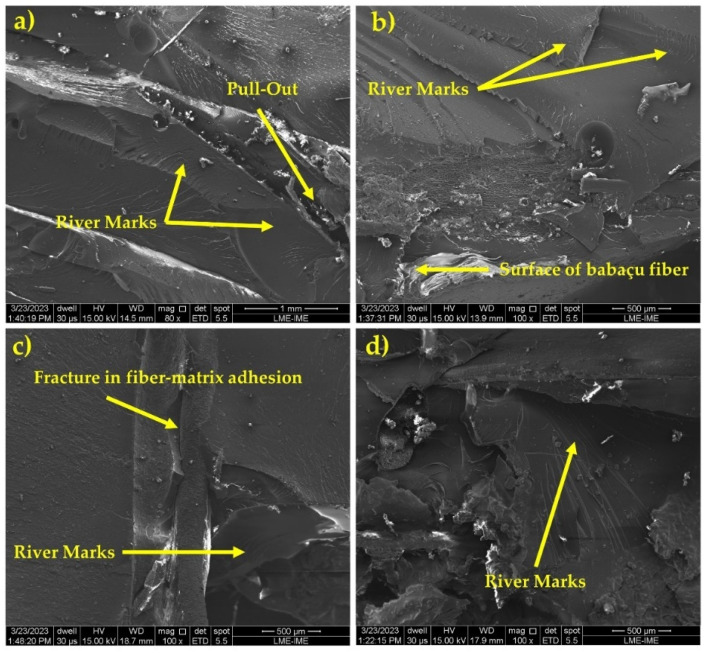
SEM microscopic appearance of the composite plate after ballistic impact: (**a**) pull-out and river marks; (**b**) river marks and the surface of babassu fiber; (**c**) fracture in fiber–matrix adhesion and river marks; (**d**) river marks.

**Figure 6 polymers-16-00913-f006:**
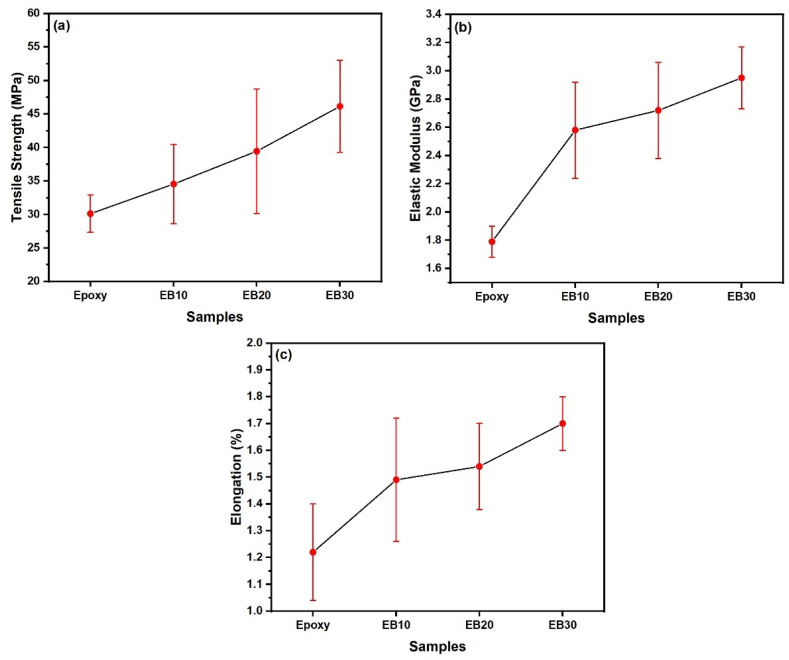
Tensile results of epoxy/babassu composites: (**a**) tensile strength; (**b**) elastic modulus; and (**c**) elongation.

**Figure 7 polymers-16-00913-f007:**
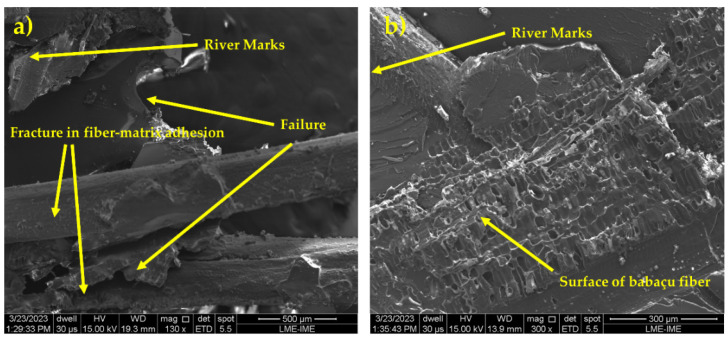
Microscopic appearance of the composite plate after the tensile test: (**a**) river marks, fracture in fiber–matrix adhesion, and failure; (**b**) river marks and the surface of babassu fiber.

**Figure 8 polymers-16-00913-f008:**
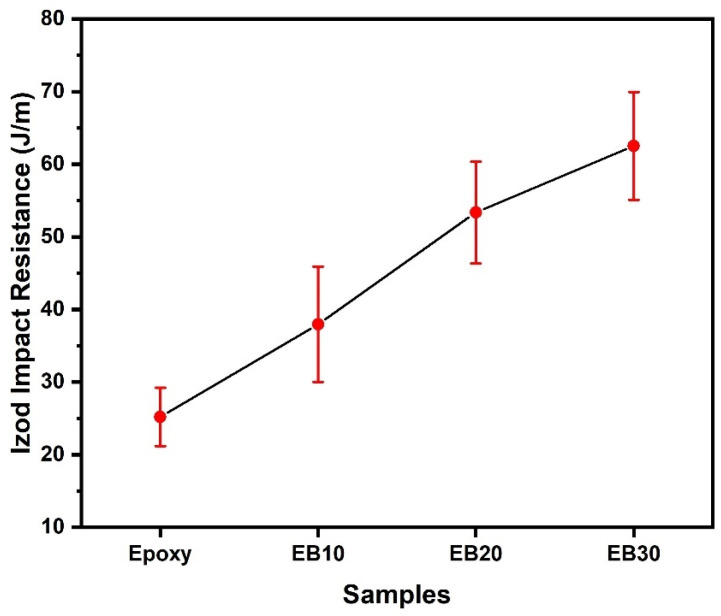
Graphical results of Izod impact tests of epoxy/babassu composites.

**Figure 9 polymers-16-00913-f009:**
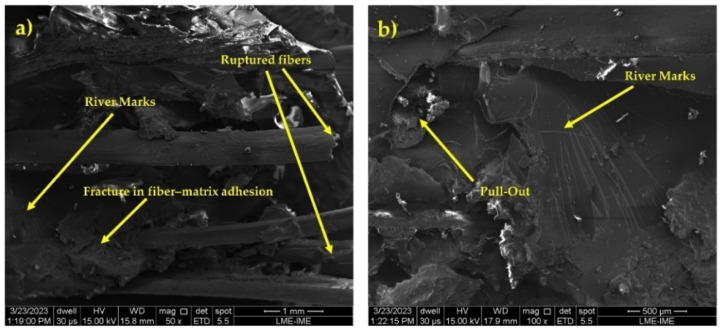
SEM microscopy appearance of the composite plate after absorption of izod impact energy: (**a**) river marks, fracture in fiber–matrix adhesion, and fiber rupture; (**b**) river marks and pull-out.

**Table 1 polymers-16-00913-t001:** Characteristics, composition, and mechanical properties of babassu fibers [26,27].

Babassu Fiber	
Diameter (mm)	0.18–0.48
Density (g/cm^3^)	0.27–0.79
Tensile Strength (MPa)	17.96–100.76
Elastic Modulus (GPa)	1.18–2.98
Crystallinity Index (%)	81.06
Microfibril Angle (°)	7.64
Moisture (%)	7.05
Lignin (%)	28.53
Hemicellulose (%)	32.34
Cellulose (%)	37.97

**Table 2 polymers-16-00913-t002:** Samples from each experimental group.

Sample Group	Composition
Epoxy	100 vol.% Epoxy
EB10	90 vol.% Epoxy + 10 vol.% Babassu fibers
EB20	80 vol.% Epoxy + 20 vol.% Babassu fibers
EB30	70 vol.% Epoxy + 30 vol.% Babassu fibers

**Table 3 polymers-16-00913-t003:** Results after ballistic test.

Samples	V_i_ (m/s)	Vr (m/s)	E_abs_ (J)	V_L_ (m/s)
Epoxy	258.90 ± 0.90	246.80 ± 3.27	44.56 ± 13.09	77.34 ± 12.75
EB10	260.18 ± 1.06	183.40 ± 13.45	202.68 ± 44.86	165.91 ± 18.71
EB20	259.44 ± 1.03	197.00 ± 13.62	163.23 ± 41.07	148.22 ± 19.47
EB30	258.90 ± 0.90	198.20 ± 14.01	156.68 ± 38.11	145.74 ± 17.19

**Table 4 polymers-16-00913-t004:** Analysis of replicates at each fiber concentration.

Number of Replicates	Fiber Concentration		
	EB10	EB20	EB30
1	208.56	213.96	218.81
2	145.77	181.98	150.24
3	218.25	146.45	158.27
4	264.60	169.68	139.53
5	176.23	104.08	116.56

**Table 5 polymers-16-00913-t005:** Variance of fiber concentration sample groups.

Sample Groups	Counting	Sum	Average	Variance
EB10	5	1013.41	202.68	2012.35
EB20	5	816.15	163.23	1686.74
EB30	5	783.41	156.68	1451.96

**Table 6 polymers-16-00913-t006:** ANOVA analysis for energy absorption results.

Sources of Variation	SQ	GL	MQ	F	*p*-Value	Critical F
Between groups	6192.29	2	3096.11	1.80	0.21	3.89
Within groups	20,604.3	12	1717.03			
Total	26,796.53	14				

**Table 7 polymers-16-00913-t007:** Grouping of groups using the Tukey method with 95% confidence.

Grouping Information Using the Tukey Method and 95% Confidence			
Group	N	Mean	Grouping
EB10	5	202.68	A
EB20	5	163.23	A
EB30	5	156.68	A

**Table 8 polymers-16-00913-t008:** Tensile test results for epoxy/babassu composites.

Sample	Tensile Strength (MPa)	Elastic Modulus (GPa)	Elongation (%)
Epoxy	30.12 ± 2.79	1.79 ± 0.11	1.22 ± 0.18
EB10	34.55 ± 5.92	2.58 ± 0.34	1.49 ± 0.23
EB20	39.44 ± 9.28	2.72 ± 0.34	1.54 ± 0.16
EB30	46.13 ± 6.87	2.95 ± 0.22	1.70 ± 0.10

**Table 9 polymers-16-00913-t009:** ANOVA analysis of tensile test results, elastic modulus, and elongation.

		Tensile			
Sources of variation	SQ	GL	MQ	F	Fc (tabulated)
Between groups	1131.53	3	377.18	8.56	2.95
Within groups	1234.2	28	44.08		
Total	2365.73	31			
		Elastic Modulus			
Sources of variation	SQ	GL	MQ	F	Fc (tabulated)
Between groups	6.04	3	2.02	27.95	2.95
Within groups	2.02	28	0.07		
Total	8.06	31			
		Elongation			
Sources of variation	SQ	GL	MQ	F	Fc (tabulated)
Between groups	0.96	3	0.32	10.60	2.95
Within groups	0.84	28	0.03		
Total	1.80	31			

**Table 10 polymers-16-00913-t010:** Tukey’s test for tensile strength tests of composites containing percentages of babassu fiber.

Tensile (HSD = 9.06)			
Group	N	Mean	Grouping
Epoxy	8	30.12	A
EB10	8	34.55	AB
EB20	8	39.44	BC
EB30	8	46.13	C
Elastic modulus (HSD = 32.63)			
Group	N	Mean	Grouping
Epoxy	8	1.79	A
EB10	8	2.58	B
EB20	8	2.72	B
EB30	8	2.95	B
Elongation (HSD = 28.19)			
Group	N	Mean	Grouping
Epoxy	8	1.22	A
EB10	8	1.49	B
EB20	8	1.54	B
EB30	8	1.70	B

**Table 11 polymers-16-00913-t011:** Results of the Izod impact tests for epoxy/babassu composites.

Sample	Izod Impact Resistance (J/m)
Epoxy	25.21 ± 4.01
EB10	37.96 ± 7.94
EB20	53.35 ± 7.02
EB30	62.53 ± 7.43

**Table 12 polymers-16-00913-t012:** ANOVA analysis of impact absorption energy results from Izod tests.

Sources of Variation	SQ	GL	MQ	F	Fc (Tabulated)
Between groups	6541.31	3	2180.44	47.46	2.95
Within groups	1286.35	28	45.91		
Total	7827.65	31			

**Table 13 polymers-16-00913-t013:** Tukey’s test for Izod energy absorption of epoxy/babassu composites.

Izod Impact Energy (HSD = 9.25)			
Group	N	Mean	Grouping
Epoxy	8	25.21	A
EB10	8	37.96	A
EB20	8	53.35	B
EB30	8	62.53	C

**Table 14 polymers-16-00913-t014:** Comparison of ballistic tests with a gauge of 0.22 carried out on composites containing babassu fiber reinforcement compared to composites containing other fibers.

Composite	Tensile Strength (MPa)	Izod Impact Resistance (J/m)	V_L_ (m/s)	E_abs_ (J)	Ref.
Epoxy	30.12 ± 2.79	25.21 ± 4.01	77.34 ± 12.75	44.56 ± 13.09	PW *
Epoxy + 10 vol.% babassu fibers	34.55 ± 5.92	37.96 ± 7.94	165.91 ± 18.71	202.68 ± 44.86	PW *
Epoxy + 20 vol.% babassu fibers	39.44 ± 9.28	53.35 ± 7.02	148.22 ± 19.47	163.23 ± 41.07	PW *
Epoxy + 30 vol.% babassu fibers	46.13 ± 6.87	62.53 ± 7.43	145.74 ± 17.19	156.68 ± 38.11	PW *
Epoxy + 40 vol.% titica vine fibers	-	58.65 ± 9.26	221.46 ± 8.10	81.02 ± 5.87	[68]
Epoxy + 30 vol.% ramie fabric + 0.5 vol.% GO **	-	-	-	130.34 ± 9.51	[69]
Epoxy + 30 vol.% kenaf fibers	-	-	-	94.81 ± 2.01	[65]
Epoxy + 30 vol.% hemp fabric	51.70 ± 9.90	134.10 ± 11.50	256.30 ± 10.50	108.50 ± 2.10	[9]
Epoxy + 30 vol.% caranan fibers	-	151.81 ± 32.04	226.56 ± 10.91	48.17 ± 8.25	[30]
Epoxy + 30 vol.% guaruman fibers	-	-	254.70 ± 12.80	105.50 ± 10.60	[35]
Epoxy + 40 vol.% tucum fibers	38.30 ± 8.10	216	-	69.60 ± 9.14	[70]

* Present work. ** Graphene oxide (GO).

## Data Availability

Data are contained within the article.

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
