# Peer review of "Mechanical and Ballistic Properties of Epoxy Composites Reinforced with Babassu Fibers (*Attalea speciosa*)"

_polymers, 2024, doi:10.3390/polym16070913_

Round 1
Reviewer 1 Report
Comments and Suggestions for Authors
This study is about investigating the mechanical and ballistic properties of epoxy composites reinforced with babassu fiber.
The incorporation of babassu fibers as reinforcement in epoxy matrix led to improvements in properties such as tensile strength, modulus, elongation, and impact energy absorption owing to the higher strength and stiffness of the lignocellulosic fibers. Ballistic testing also showed over 3.5 times increase in energy absorption for the composites, with the 10% composite maintaining integrity after multiple shots. The study has been well conducted. However, the presentation of data and results require further improvement. The authors have included many raw data which in my opinion can be omitted.
- The selection of 10-30 vol% of the babassu fiber should be justified.
- Most of the results are presented in table form. For better presentation, they should be plotted in graph.
Table 7 & 10: The authors should not include the raw data.
- Natural fiber-reinforced epoxy composites have been well studied. Therefore, comparison on the physical and mechanical properties of the composites produced in this study should be provided.
- Chemical composition and dimension (diameter & length) of the babassu fibers should be provided.
The manuscript requires thorough revision before it can be considered for publication in Polymers.
Comments on the Quality of English LanguageThe manuscript is still in very raw form. There are many raw data which should be excluded from the manuscript.
Author Response
Dear Reviewer,
We thank for their careful review of our manuscript and valuable comments and suggestions, which have allowed us to improve the quality of the paper.

Reviewer 2 Report
Comments and Suggestions for Authors
Author Response

(The authors gave the same response as above.)

Reviewer 3 Report
Comments and Suggestions for Authors
In this paper, the ballistic behaviors of a typical Brazilian natural fiber reinforced epoxy composite was studied. Altough this kind of natural fiber was not investigated before, there was no any new scientific findings, such as mechanical properties and failure mechiams, in this paper. Therefore, I recommend Reject.
Author Response

(The authors gave the same response as above.)

Reviewer 4 Report
Comments and Suggestions for Authors
The paper is interesting, It needs some revisions before processing further.
1. Since babassu is a new type of fibers, a more details ablout the plant, the dimensions, microscopy (SEM) of fiber crosssection, the fiber mechanical and other physical properties should be presented in detail.
2. A comparative account of this fiber with pther fibers used in similar composites e.g., jute, sisal, coir etc. shpuld be included.
3. The cellulose content, moisture regain are also important propertieswhich should be included to the table of comparison.
4. If only the tensile properties are studied, then the title should be "Tensile and ballistic properties..............." instead of "Mechanical and ballistic properties............".
5. Why the results are not presented in terms of stress-strain curves and only in tables? Further the variability should be given. In stead of showing the raw data of 8 samples, it is better to present barchart with error bars.
6. Why at all ballistic properties are studied? What is the application area of such composites where ballistic performance is important should be justified clearly.
7. The abstract and conclusion should be modified accordingly.
Comments on the Quality of English LanguageModerate
Author Response

(The authors gave the same response as above.)

Round 2
Reviewer 1 Report
Comments and Suggestions for Authors
The authors have revised the manuscript accordingly. Therefore, the manuscript is now acceptable for publication.
Reviewer 4 Report
Comments and Suggestions for Authors
Can be accepted
Comments on the Quality of English LanguageMinor editing